# An Assessment of Energy and Groundwater Consumption of Textile Dyeing Mills in Bangladesh and Minimization of Environmental Impacts via Long-Term Key Performance Indicators (KPI) Baseline

Abdullah Al Mamun [1], Koushik Kumar Bormon [2], Mst Nigar Sultana Rasu [3], Amit Talukder [4], Charles Freeman [4,*], Reuben Burch [1,5] and Harish Chander [5,6]

1    Department of Industrial and Systems Engineering, Mississippi State University, Starkville, MS 39762, USA
2    Department of Hydro Science and Engineering, Technical University of Dresden, 01062 Dresden, Germany
3    Hohenstein Institute Bangladesh, Dhaka 1213, Bangladesh
4    Department of Human Sciences, Mississippi State University, Starkville, MS 39762, USA
5    Human Factors and Athlete Engineering, Center for Advanced Vehicular Systems, Mississippi State University, Starkville, MS 39762, USA
6    Neuromechanics Laboratory, Department of Kinesiology, Mississippi State University, Starkville, MS 39762, USA
*    Correspondence: cf617@msstate.edu

**Abstract:** Bangladesh's ready-made garment sectors have evolved to increase market share in the global textile supply chain. Textile sectors heavily rely on energy and groundwater consumption during production; mainly, textile dyeing mills contribute to the carbon footprint and water footprint impact to the environment. Textile dyeing mills have become one of the major industries responsible for the continuous depletion of groundwater levels and severe water pollution to the environment. Reduction of long-term key performance indicators (KPI) can be set to a baseline by reducing energy and groundwater consumption in textile dyeing mills. This study has analyzed the energy and groundwater consumption trend based on 15 textile dyeing mills in Bangladesh in 2019. The average dyed fabric production of 15 textile dyeing mills in 2019 was 7602.88 tons by consuming electricity and groundwater, and discharging treated effluent wastewater to the environment, in the amounts of 17,689.43 MWh, 961.26 million liters, and 640.24 million liters, respectively. The average KPI of treated effluent discharged wastewater was 97.27 L/kg, and energy consumption was 2.58 kWh/kg. Considering yearly 5% reduction strategies of groundwater and energy consumption for each factory could save around 355.43 million liters of water and 6540.68 MWh of electricity in 10 years (equivalent to 4167.08-ton $CO_2$ emission).

**Keywords:** effluent treatment; energy and water footprint; groundwater level; key performance indicator; heavy metals discharge

## 1. Introduction

Energy and water play a vital role in the world's textile supply chain. The product lifecycle of a ready-made garment is related to energy and water consumption (EWC) that comprises several phases: utilization of agricultural machinery driven by fossil fuel and water usage during cotton cultivation; EWC in textile production: spinning, weaving, dyeing/finishing, and apparel manufacturing; logistics and transportation of ready-made garments which contributes to energy consumption; personal use of washing machines that require a significant amount of water and energy [1–4]. The environmental impact of the textile supply chain is widespread; for example, it annually contributes 1.7 billion tons of $CO_2$ emissions, which is around 10% of global greenhouse gas (GHG) exposure [1]. The

textile supply chain also consumes 1.5 trillion liters of water each year, which is responsible for 20% of industrial water pollution [5].

Bangladesh is the second-largest exporter of global ready-made garments (RMG), followed by China [6]. In 2019, the total export value of RMG was 34.13 billion US$, contributing 84.21% of Bangladesh's total export value. In Bangladesh, the RMG sector has evolved to expand its global market share and increase its export value by approximately 63.40% (from 2009 to 2019) [7]. However, the RMG sectors of Bangladesh heavily rely on energy and groundwater consumption during the production process, contributing to carbon footprint (CFP) and wastewater discharge to the environment, respectively. Therefore, EWC in the ready-made garments sector in Bangladesh has become a significant concern for environmental sustainability. However, scarcity of sustainable water may hamper the continuous growth of the RMG sector in Bangladesh, mainly groundwater, the largest and only water source for the entire textile dyeing industry [8]. Due to the self-extraction of unpriced groundwater in most factories, textiles have become one of the major industries responsible for the continuous depletion of groundwater levels and water pollution [9]. In most textile factories, the usage of extracted groundwater is inefficient, and the amount of attenuation is insignificant. In 2015, 1700 textile dyeing mills in Bangladesh consumed approximately 1500 billion liters of groundwater. After groundwater usage by textile dyeing mills, they discharge treated effluent wastewater into the environment, causing extreme water pollution and groundwater depletion [10]. In addition, the surface water of the nearby rivers and water canals has been contaminated by this discharged wastewater with harmful fragments of dyes and chemicals, ultimately affecting aquatic ecology and agriculture.

Over the past two decades, groundwater decline has significantly threatened the area in and around Dhaka city and adjacent industrial zones [11]. The extraction is more than the recharge of aquifers, causing the deterioration of groundwater levels [8,12]. With the depletion of the groundwater level, the energy cost for groundwater extraction will also impact production costs in the RMG sector. Therefore, it is high time to tackle this alarming situation to save our environment and the RMG industry. Addressing this issue, establishing a benchmark of key performance indicators (KPI) of energy, groundwater and treated effluent wastewater based on the amount of dyed fabric production will help the sustainable environment performance index.

### 1.1. Water and Energy Consumption in Textile Dyeing Mills

By the year 2050, the world's population will increase around 35%, significantly increasing textile production and consumption, driving a significant increase in energy and water consumption, ultimately leading to environmental pollution [4]. In addition, the textile industry requires an intensive amount of water, which significantly strains global water resources. As a result, the textile industry is accounted the worst polluter of clean water, followed by agriculture. At the same time, there are significant concerns about wet textile processing consuming a massive amount of freshwater, discharging wastewater and polluting the ecosystem [13]. For example, in 2016, the Chinese textile industry (consisting of 50,000 textile factories) consumed approximately 3000 billion liters of freshwater [2,6]. According to the Turkish Statistical Institute, the textile industry is responsible for 191.5 billion liters of water consumption, the second-largest industry in the manufacturing sector [14]. On average, approximately 2500–3000 L of water are required to manufacture a cotton t-shirt. Moreover, a substantial amount of water consumption is associated with cotton cultivation, followed by wet processing [2]. In addition, conventional textile dyeing and finishing process require approximately 1.5 million liters of water for every ton of textile processing [13]. Researchers measure specific water consumption (SWC) (treated/groundwater) as usage per mass of the product [9]. For example, various investigations showed that in the wet textile process, on average, 200–400 L of water were consumed for dyeing 1 kg of fabric [15–17]. In the meantime, SWC usage in the Turkish textile sector varies from 20 L/kg to 230 L/kg [18]. Therefore, SWC can help

promote water footprint awareness and set a benchmark index regarding environmental sustainability. Similarly, yarn spinning, and wet processing consume significant electricity from the national grid and captive power generators, using fossil fuel and natural gas for the textile industry. Therefore, manufacturers measure specific energy consumption (SEC) in wet textile processing as a ratio of electricity consumption for dyeing 1 kg fabric (kWh/kg) [19]. Generally, the SEC of a textile dyeing mill plays a vital role in monitoring electricity usage versus production calculation. SEC is convertible to a carbon footprint based on country-wise "emission factors." In Bangladesh, 0.64 kgCO$_2$ contributes to the environment, equivalent to generating 1 kWh of electricity [20]. An investigation found that an average SEC in Turkish textile wet processing required 3.4 kWh/kg dyed fabric [21]. Conserving energy and water consumption will help mitigate air and water pollution, which will also be part of a more environmentally friendly production process.

### 1.2. Impact of Discharged Wastewater on the Environment

Discharged treated effluent wastewater contains intense color, inorganic finishing agents, surfactant, chlorine compounds, high chemical oxygen demand (COD) and bio-chemical oxygen demand (BOD) amounts, and heavy metals [22]. The investigation also showed that wet textile processes, including bleaching, dyeing, printing, and finishing, use 3600 dyes and 8000 chemicals [23]. Therefore, effluent treatment costs may account for 5% of total production costs [6]. However, the conventional effluent treatment method is unsuitable for purifying many toxic and bio-degradable compounds in wastewater [14,17]. Many of these dyes and chemicals account for the direct and indirect causes of water pollution, soil contamination, and threats to aquatic life [24]. An estimation showed that textile effluent discharge was around 280,000 tons of textile dyes annually around the globe. In addition, the discharged treated effluent wastewater temperature is higher (65 °C) than regular water, reducing the dissolved oxygen level of normal water and leading to an imbalance of biodiversity [24]. Due to this, China is facing one of the worst water pollution scenarios, which has happened because 70% of China's rivers, lakes, and reservoirs have already been contaminated mainly by textile industries [2].

In Bangladesh, a massive amount of discharged wastewater from textiles and effluents has already altered the aquatic ecosystem's chemical and physical properties. This alteration of the typical marine environment has impacted humans, livestock, the fish population, and biodiversity [25,26]. Moreover, untreated textile wastewater also has a severely harmful effect on groundwater quality. The location of textile industries is clustered within a range of 60 km in greater Dhaka industrial zones and their vicinity. The region includes Narayangonj, Gazipur, some of Mymensingh, and Narsingdi, where rivers and water canals near these zones are being polluted by discharged wastewater from textile dyeing industries [27]. Major affected rivers in these textile industrial zones include Buriganga, Shitalakkhya, Turag, and Dhaleshwari (Figure 1) [28]. Addressing this dire condition of water pollution, implementing advanced technologies, and cleaner production strategies may help reduce water consumption and effluent volume from textile dyeing industries.

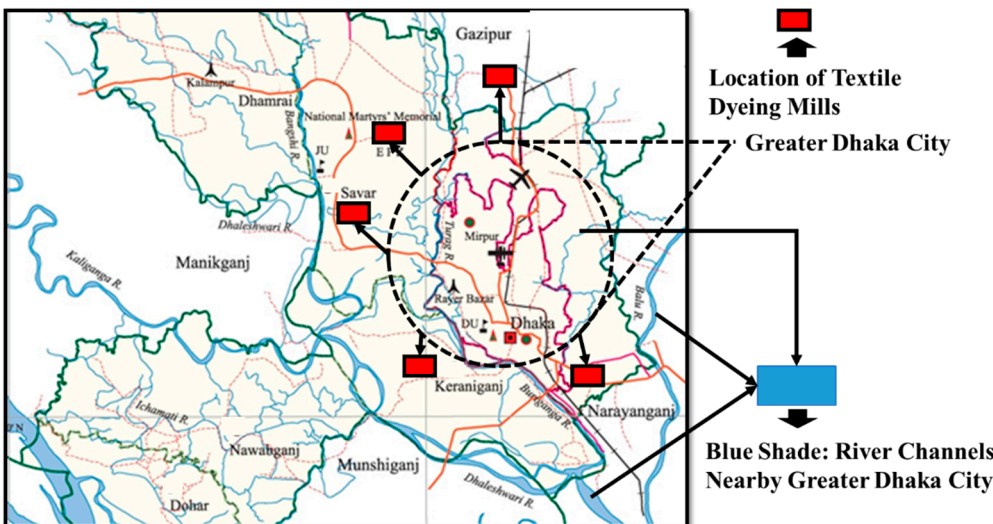

**Figure 1.** River water pollution by industrial waste in adjacent rivers of Dhaka city. Adapted from [28].

*1.3. Groundwater Level Depletion*

Water pollution and scarcity of water resources have become severe problems due to a lack of wastewater treatment and water abuse [6]. Freshwater consumption in the wet textile process has also become a significant concern for those countries facing water shortages or those facing it in the near future. For example, textile dyeing mills in Bangladesh utilize a considerable amount of groundwater. As a result, the decrease of groundwater and increasing surface water pollution coincide. Industrial effluents from textile dyeing industries are destroying nearby surface water resources. Currently, water treatment of Shitalakkhya river water by DWASA (Dhaka Water and Sewerage Authority) meets around 22% of the 2.3 billion liters of daily water demand in Dhaka City, with the remaining needs met through underground water resources [29]. Generally, extracted groundwater needs significantly fewer water treatment procedures, while surface water requires various treatment processes that involve substantial investment costs for drinking, domestic and industrial purposes. However, an investigation showed that Dhaka's groundwater level has dropped by 200 feet in the last 50 years, and this trend continues at a high rate [30]. Consequently, large volumes of groundwater extracted by the textile dyeing industries threaten the quality and quantity of drinking water accessible to the residents of Dhaka City.

This study aims to examine the current trend of CFP, effluent discharge (wastewater), and groundwater depletion levels based on textile dyeing mills. Long-term improvement of these trends based on the amount of dyed fabric production will help maintain a sustainable environment in Bangladesh.

## 2. Materials and Methods

*2.1. Study Approach*

Figure 2 represents the overall study approach of collecting data, processing data, and analyzing KPI based on production data from the wet processing unit, groundwater extraction volume, amount of discharged wastewater from treated effluent, energy consumption from the national grid, and captive power generation and water & carbon footprint. Data collection, KPI analysis, and recommendations are demonstrated in Sections 2.2, 3.1 and 3.2, respectively. The recommendation section comprises three key points: making a database for future reference, setting goals for a yearly KPI% reduction, and introducing the best available technology to increase productivity and reduce water and energy consumption.

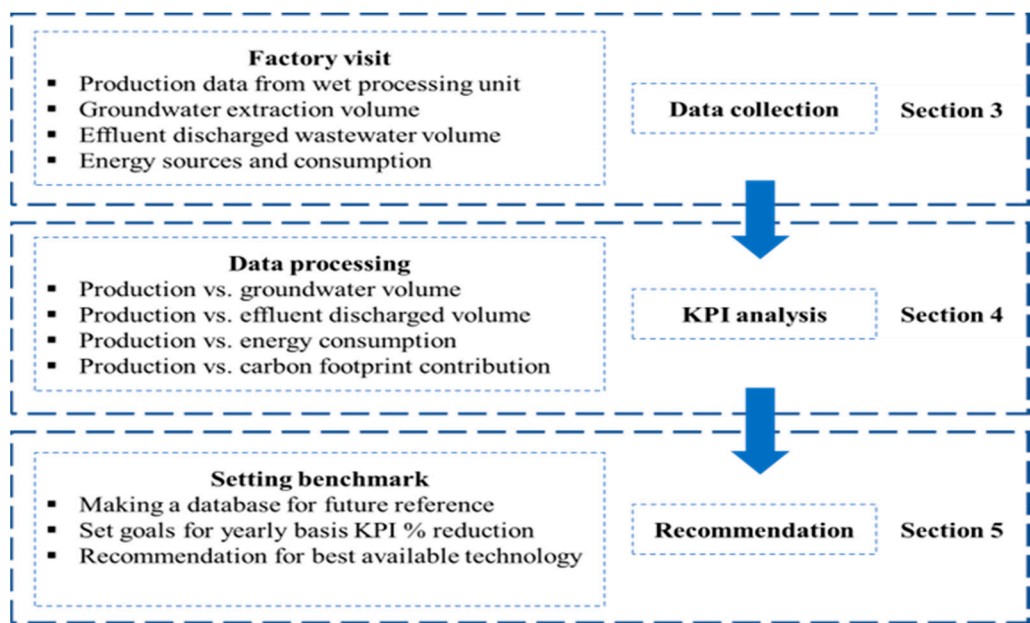

**Figure 2.** Work procedure of data processing and analyzing KPIs.

*2.2. Data Collection*

Factory management collected and provided data to the authors from 15 textile factories based in 2019. Data was collected based on dyed fabric amounts from 15 textile factories and associated extracted groundwater, consumed energy, and discharged wastewater. The factory distances from Dhaka city's center range from 20 km to 60 km (Figure 3). In addition, researchers collected secondary data from journal papers, survey reports, international conference papers, newspapers, and textile magazine articles to corroborate data from the 15 participant sites.

### Factory Distance from Center of Dhaka City (Km)

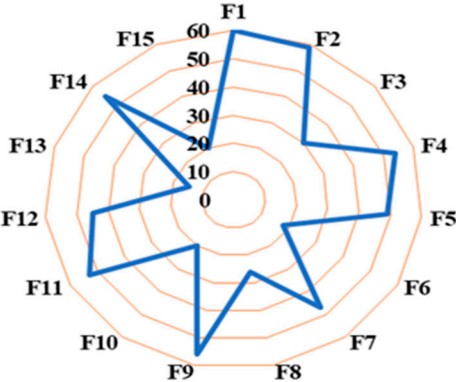

**Figure 3.** List of factory distances from the center of Dhaka city.

Groundwater Demand and Wastewater Discharge into the Environment

Equations (1)–(3) demonstrate the KPI of groundwater extraction, effluent discharged wastewater, and water loss in the process, respectively. The annual KPI of groundwater is a ratio between the extracted groundwater amount (L) and total dyed fabric amount (kg). Similarly, the yearly KPI of wastewater is the ratio between total treated effluent discharged wastewater and the total dyed fabric amount (kg) in the same year.

$$\text{KPI}_{\text{Groundwater}} = \frac{\sum \text{ Extracted groundwater (Liter)}}{\sum \text{ Dyed fabric amount (kg)}} \qquad (1)$$

$$\text{KPI}_{\text{Wastewater}} = \frac{\sum \text{Discharged wastewater (Liter)}}{\sum \text{Dyed fabric amount (kg)}} \tag{2}$$

Equation (3) shows that the KPI of water loss in the process is measured based on Equations (1) and (2) calculates the difference between groundwater and wastewater KPIs.

$$\text{KPI}_{\text{Water loss in the process}} = \text{KPI}_{\text{Groundwater}} - \text{KPI}_{\text{Wastewater}} \tag{3}$$

### 2.3. Data Collected from Selected Dyeing Mills

Table 1 demonstrates collected data from 15 textile dyeing mills in 2019. Data were collected in each factory based on the annual dyed fabric amount, extracted groundwater volume, the treated effluent wastewater discharge volume, and electricity consumption. Electricity supply from the national grid source and electricity from captive power generation using fossil fuel and natural gas lines determined total electricity consumption. In addition, researchers tracked groundwater extraction and effluent discharged wastewater using a water outlet flow meter.

**Table 1.** Resource Consumption Data Collected from 15 Textile Dyeing Mills.

| Factory | Total Production (kg) | Total Electricity Use (kWh) | Extracted Groundwater (in Million Liters) | Discharged Wastewater (in Million Liters) |
|---|---|---|---|---|
| F1 | 4,330,515.00 | 15,125,711.00 | 744.05 | 624.19 |
| F2 | 9,088,174.00 | 20,034,081.00 | 1455.87 | 865.66 |
| F3 | 4,413,619.00 | 6,673,371.00 | 687.66 | 340.77 |
| F4 | 3,639,554.00 | 7,566,789.00 | 533.58 | 501.68 |
| F5 | 2,409,076.00 | 8,099,029.00 | 477.45 | 372.53 |
| F6 | 4,083,918.00 | 13,265,449.00 | 703.73 | 481.55 |
| F7 | 11,193,569.00 | 21,806,157.00 | 1297.86 | 922.51 |
| F8 | 4,355,228.00 | 6,595,024.00 | 470.36 | 270.21 |
| F9 | 4,049,474.00 | 6,992,793.00 | 401.49 | 350.37 |
| F10 | 10,702,135.00 | 6,5546454.00 | 1302.71 | 851.50 |
| F11 | 5,223,919.00 | 9,699,231.00 | 652.66 | 358.01 |
| F12 | 41,530,362.00 | 57,402,492.00 | 4526.32 | 2815.04 |
| F13 | 2,553,747.00 | 4,194,361.00 | 356.67 | 260.12 |
| F14 | 2,895,710.00 | 4,574,060.00 | 371.86 | 281.81 |
| F15 | 3,574,246.00 | 17,766,474.00 | 436.65 | 307.67 |

Energy Consumption and Carbon Footprint

Carbon footprint (CFP) is a broadly used tool for monitoring global climate change. CFP's impact on the environment is attributed to the emission of greenhouse gases (GHGs) such as $CO_2$, $N_2O$, hydrofluorocarbons (HFCs), perfluorocarbons (PFCs), and sulfur hexafluoride (SF6) [4,31]. CFP is measured as grams of $CO_2$ equivalent to generating per kilowatt-hour of electricity ($gCO_2eq/kWh$) utilizing hydrocarbon-containing fossil fuel [32,33]. The emission factor (EF) varies from country to country, which depends on the resource utilization of fossil fuels. Table 2 shows country-wise emissions per kWh of electricity depending on carbon heat generation [20]. For example, country-wise emission per kWh in Bangladesh is 0.6371 kg-$CO_2$, while $KgCH_4$ and $KgN_2O$'s environmental contribution is insignificant. Based on Table 2, India is the highest contributor per kWh equivalent $CO_2$ emission to the environment, whereas Cambodia and China stood in the second and third positions for $CO_2$ emission. Additionally, Table 2 can also compare country-wise carbon footprint impact broadly related to textile dyeing production in the global textile supply chain. More elaborately, a fair KPI can be reached based on a particular time of textile dyed fabric amount and electricity consumption using the unit kWh/kg. For instance, 1 kWh of electricity production in Bangladesh contributes 0.6371 kg $CO_2$ and

converts kWh/Kg to 0.6371 kg $CO_2$/kg. Similarly, this unit can be presented as 0.9746 kg $CO_2$/kg in China. A list of country-wise emissions factors is adapted from [20]

$$\text{Emission of GHGs=Energy Consumption (EC)} \times \text{Emission Factor (EF)} \tag{4}$$

**Table 2.** Country-wise emissions per kWh of electricity generated [20].

| Country | kgCO$_2$/kWh | kgCH$_4$/kWh | kgN$_2$O/kWh |
|---|---|---|---|
| Bangladesh | 0.6371 | 0.00001236 | 0.00000191 |
| China | 0.9746 | 0.00001047 | 0.00001521 |
| Cambodia | 1.1708 | 0.00004638 | 0.00000928 |
| India | 1.3332 | 0.00001552 | 0.00002011 |
| Pakistan | 0.4734 | 0.00001384 | 0.00000243 |
| Vietnam | 0.4668 | 0.00000705 | 0.00000420 |
| Sri Lanka | 0.4172 | 0.00001644 | 0.00000329 |

Equation (5) shows the emission of GHG based on energy consumption and country-wise emission factor. Equation (5) represents energy consumption based on national grid supply and captive power generation. Calculations use solar energy as a negative emission factor. However, the amount of solar energy is insignificant compared to grid electricity and a captive power source.

$$\text{EC (kWh)} = \sum (\text{Grid electricity} + \text{Captive power generation}) - \sum \text{Solar energy} \tag{5}$$

Equation (6) represents the KPI of energy consumption based on a ratio of total electricity consumption (kWh) and total dyed fabric amount (kg). Equation (7) shows the KPI of $CO_2$ emission contribution to the environment based on a ratio of total electricity consumption (kWh) times per kWh equivalent emission factor to total dyed fabric amount (kg).

$$KPI_{Energy} = \frac{\sum \text{EC (kWh)}}{\sum \text{Dyed fabric amount (kg)}} \tag{6}$$

$$KPI_{CO_2} = \frac{\sum \text{EC(kWh)} \times \text{Emission Factor(EF)}}{\sum \text{Dyed fabric amount (kg)}} \tag{7}$$

## 3. Results

### 3.1. Production vs. Energy Consumption

Figure 4 demonstrates the KPI of 15 factories based on energy consumption and production data for 2019. Based on Table 2 and Equation (6), the annual energy consumption KPI is the total dyed fabric ratio in a year. As a result, researchers calculated the mean KPI of 15 factories to be 2.58 kWh/kg, where the maximum and minimum KPI was 6.12 kWh/kg and 1.38 kWh/kg, respectively.

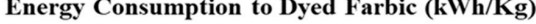

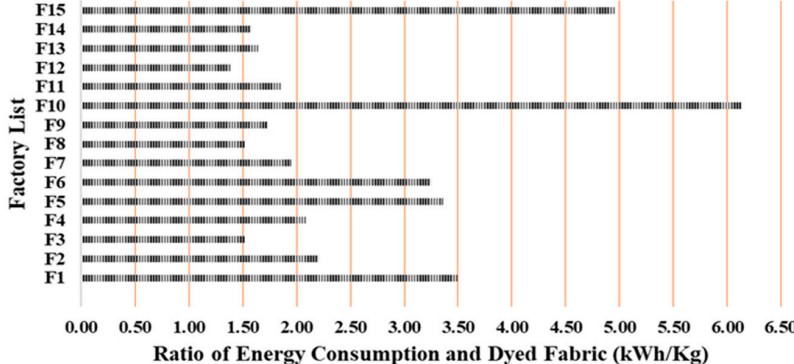

**Figure 4.** The ratio of energy consumption and dyed fabric (kWh/kg).

### 3.2. Carbon Footprint Contribution to kWh Electricity and Dyed Fabric

Figure 5 represents carbon footprint contribution equivalent per kWh electricity consumption and dyed fabric amount of 15 factories. Based on Table 1 and Equation (7), the KPI of carbon emission contribution uses a ratio of Bangladesh's yearly energy consumption multiplied by emission factors (Table 2) to the total dyed fabric in a year. As a result, the average carbon emission KPI of 15 factories was 1.64 kg-$CO_2$/kg, where the maximum and minimum KPI was 3.90 kg-$CO_2$/kg and 0.88 kg-$CO_2$/kg, respectively.

**Figure 5.** The ratio of energy consumption and dyed fabric (kWh/kg).

### 3.3. Groundwater vs. Discharged Wastewater Comparison

Figure 6 compares groundwater versus effluent discharged wastewater based on dyed fabric amounts of 15 textile dyeing industries. The maximum, minimum, and average KPI of extracted groundwater were 198.20 L/kg, 99.15 L/kg, and 138.26 L/kg, respectively. Similarly, the maximum, minimum, and average KPI of effluent-treated discharged wastewater was 154.64 L/kg, 62.04 L/kg, and 97.27 L/kg, respectively. Finally, researchers calculated the KPI of water loss in the process by taking the difference between groundwater and effluent discharged wastewater KPI, where maximum, minimum, and average KPI differences were found at 79.0 L/kg, 9.0 L/kg, and 41.0 L/kg, respectively.

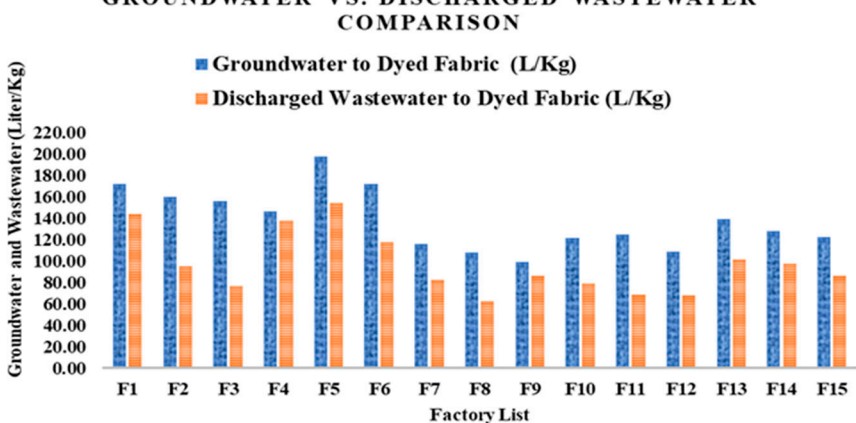

**Figure 6.** Groundwater vs. effluent discharged wastewater comparison based on dyed fabric amount (kg).

### 3.4. Heavy Metal Discharge with Treated Wastewater

Figure 7 shows heavy metals released into the environment with effluent-treated wastewater from 15 textile dyeing mills. According to an effluent-treated wastewater analysis report from 15 textile dyeing mills, results detected nine heavy metals (boron, manganese, chromium, zinc, copper, nickel, cobalt, antimony, and lead). Zinc (Zn) was detected in 13 out of 15 factories, while cobalt (Co) and boron (B) were the lowest traced

heavy metals found in only two factories. The trace of heavy metals was incorporated from the factory-wise effluent-treated wastewater analysis reports from third-party laboratories.

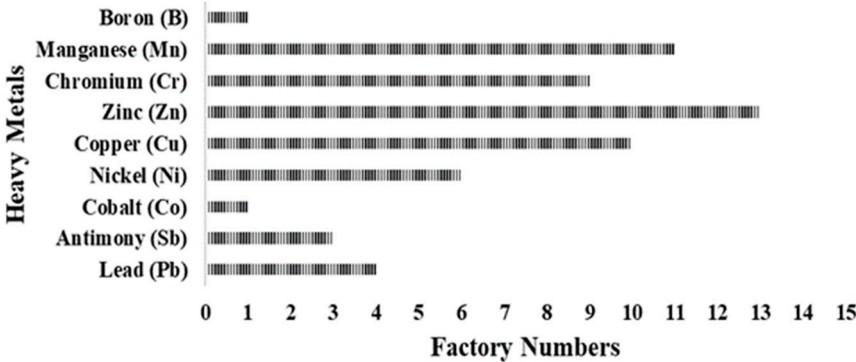

**Figure 7.** Heavy metal released with effluent-treated wastewater.

*3.5. COD and BOD Amount in Effluent Discharged Wastewater*

Figure 8 represents the COD, and BOD amount in effluent discharged wastewater according to the factory-provided wastewater analysis report. Using a wastewater analysis report, researchers found maximum, minimum, and mean COD in the wastewater at 216 mg/L, 28 mg/L, and 88 mg/L, respectively. Similarly, the maximum, minimum, and average BOD amounts were traced at 44 mg/L, 4 mg/L, and 21.8 mg/L, respectively.

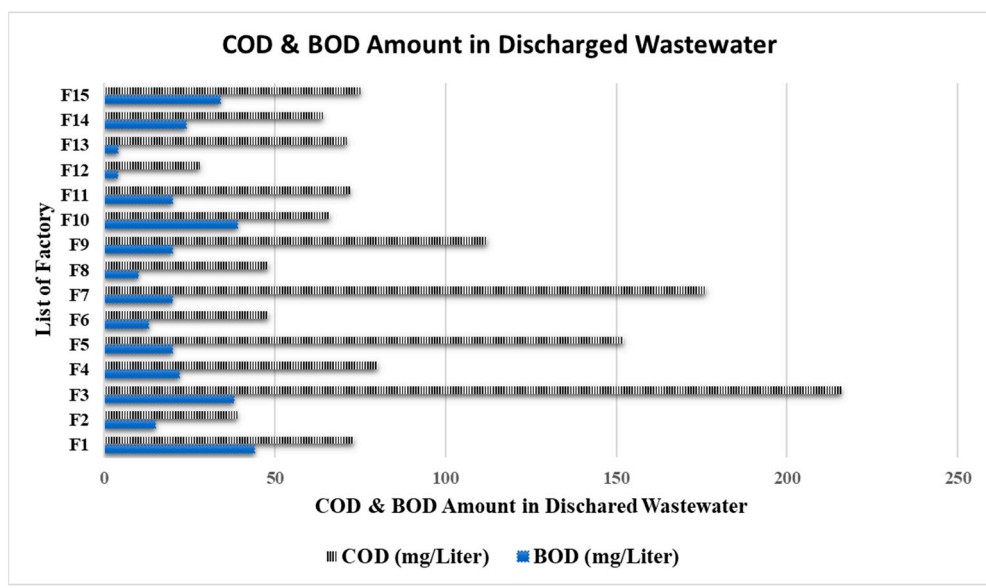

**Figure 8.** COD & BOD amount in effluent discharged wastewater.

## 4. Discussion

*4.1. Yearly Basis KPI% Reduction Approach and Potential Saving*

Figure 9 demonstrates a potential groundwater saving (yearly 5% reduction) approach based on an average groundwater extraction amount of 961.26 million liters per factory. Using 5% reduction strategies of groundwater for each factory can save around 355.43 million liters in 10 years. Figure 10 shows a potential energy-saving approach for a single factory in 10 years. Similarly, a 5% reduction strategy of average energy consumption (17,689.43 MWh) for a single factory can save 6540.68 MWh of electricity in 10 years, equivalent to 4167.08 tons of $CO_2$ emission reduction to the environment [31].

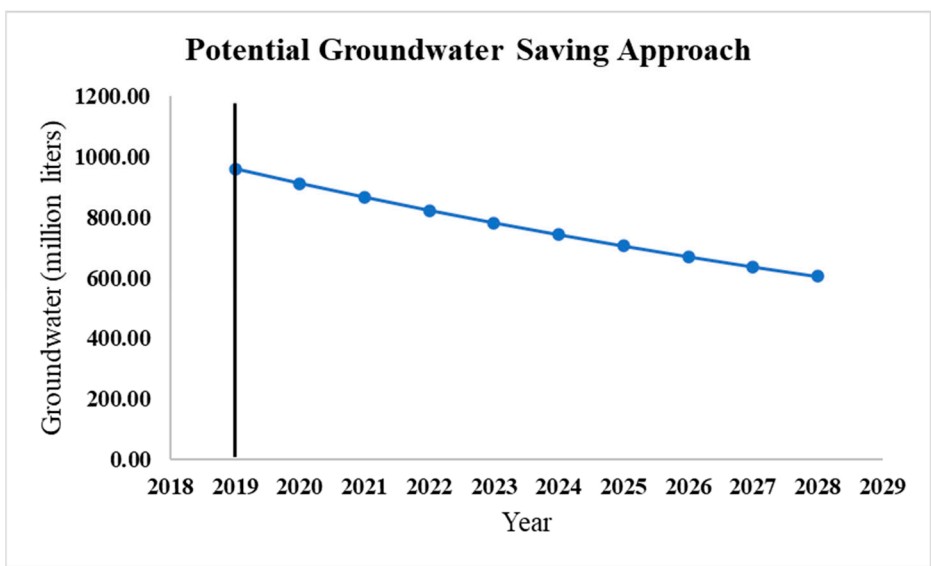

**Figure 9.** Potential groundwater saving approach in 2019.

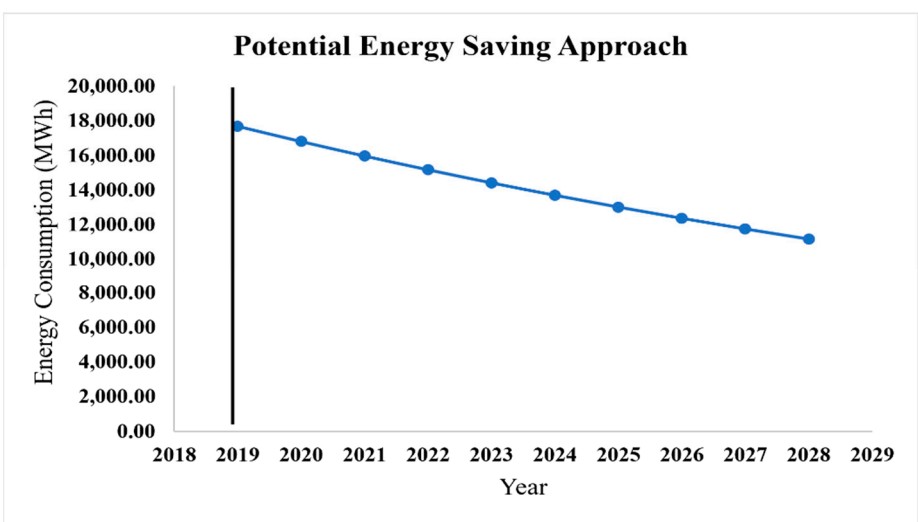

**Figure 10.** Potential energy saving approach in 2019.

### 4.2. Recommendations

Factories can adopt different strategies to minimize water and energy losses to save a potential amount of water and energy without significant investment. However, new machinery and equipment setup often require a considerable investment. Some recommendations are highlighted in Sections 4.2.1–4.2.3.

### 4.2.1. Best Available Techniques for Potential Water-saving Approaches

- Process-wise and machine-wise water consumption should be monitored for individual dyeing machine water consumption and take initiatives where water consumption is comparatively high;
- Use water-efficient machinery and equipment, for instance, substituting a high liquor ratio dyeing machine with a low liquor ratio [13];
- Ensure optimum condensate recovery from all sections by installing steam traps and condensate transfer pump to reuse as boiler feedwater;
- Reuse of effluent-treated wastewater in cleaning the empty chemical drums, printing screens and production floor, car washing and toilet flushing [14];

- Rainwater is much purer than groundwater and can be used in the production process without passing through the WTP (Water Treatment Plant), which is economically suitable and environmentally friendly;
- Prevent all leakages in the waterline and use a trigger nozzle in hose pipes to avoid the excessive flow of water;
- For fabric washing purposes, counter-current rinsing should be followed. Pretreatment washing of the dyed fabric should be conducted according to the requirement to avoid excess washing. Additionally, avoiding the excessive washing of machines;
- Adopting digital printing instead of a standard printing system where chemical wastage is minimal and requires less water;
- Provision of recovering salt from used liquor, which is ultimately drained to ETP, increasing treatment cost and using acid to neutralize the high amount of alkaline effluent.

### 4.2.2. Best Available Techniques for Potential Energy-Saving Approaches

- Substitute manual blowdown of boilers with an auto blowdown system to save energy;
- The concept of smart lighting involves utilizing natural light from the sun. Smart lighting is also a good initiative that minimizes and saves light by allowing the proper place lighting;
- Proper insulation of all steam valves & flanges to avoid heat loss
- Maintain proper air and fuel ration in boiler through oxygen tuning/oxygen analyzer to reduce excessive natural gas consumption;
- Installation of exhaust gas boiler and heat recovery from flue gas by installing an economizer;
- Performing regular leakage tests and monitoring the leakage level of compressed air lines.

### 4.2.3. Factory Management Initiatives

- Employee and worker training on water usage also plays a significant role [14]. Conveying the environmental impact and the growing consciousness of illiterate or less-educated workers is very important. The feasibility of waterless dyeing with $CO_2$ or plasma processing should be investigated as a pilot project basis in Bangladesh as soon as possible. With modern techniques and solid economic background, some countries are introducing absolute recycling of water through the zero liquid discharge (ZLD) plant, which could be the ultimate solution for toxic wastewater. As Dhaka's groundwater level is significantly declining, some researchers have suggested recharging the groundwater artificially [11].
- These approaches could be taken to minimize water and energy without significant investment. However, this study has analyzed the energy and groundwater consumption trend based on 15 textile dyeing mills in Bangladesh in 2019. The article was set up as a critical review of the failure criteria that guide the selection of the most suitable criterion for the chosen case study. Long-term key performance indicator (KPI) reduction is set to a baseline by reducing energy and groundwater consumption in textile dyeing mills. The overall calculation can vary by location of textile dyeing mills worldwide, the number of textile dyeing mills, and the timelines. This case study was limited to energy and groundwater consumption trends in textile dyeing mills in Bangladesh. Future recommendations of this study could be expanded to other textile regions in Bangladesh.

## 5. Conclusions

Bangladesh is the second-largest exporter of global RMG, followed by China, and this RMG sector has evolved in growing global market share and increasing its export value by approximately 63.40% (from 2009 to 2019). These RMG sectors heavily rely on energy and groundwater consumption during the production process, contributing to carbon footprint and wastewater discharge to the environment. With the shortage of groundwater levels,

the energy cost for groundwater extraction will also impact production costs in the RMG sector. However, soon scarcity of sustainable water may hamper the continuous growth of the RMG sector in Bangladesh, mainly relating to use of groundwater. Over the past decades, groundwater decline has been a major threat to Greater Dhaka city and adjacent industrial zones. Meanwhile, the extraction is more than the recharge of aquifers, causing the deterioration of groundwater levels. After comparing dyed fabric amounts of 15 textile dyeing mills and energy consumption, the average KPI of 15 factories was found to be 2.58 kWh/kg. Therefore, on average, COD and BOD in effluent discharged wastewater of 15 factories were 88 mg/L and 21.8 mg/L, respectively. A yearly 5% reduction strategy of groundwater and energy consumption for each factory can save around 355.43 million liters of groundwater and 6540.68 MWh of electricity in the next ten years in Bangladesh (equivalent to 4167.08-ton $CO_2$ emission). Therefore, without hampering global demands, this saved water and energy could help us survive more sustainably in the future.

**Author Contributions:** Conceptualization, A.A.M., K.K.B. and M.N.S.R.; methodology, A.A.M., K.K.B. and M.N.S.R.; investigation, A.A.M., K.K.B. and M.N.S.R.; visualization, A.A.M., K.K.B. and M.N.S.R.; data curation A.A.M., K.K.B. and M.N.S.R.; writing—original draft preparation, A.A.M., K.K.B., M.N.S.R. and A.T.; writing—review and editing, A.T., C.F., R.B. and H.C.; supervision, C.F., R.B. and H.C. All authors have read and agreed to the published version of the manuscript.

**Funding:** This research received no external funding.

**Institutional Review Board Statement:** Not applicable.

**Informed Consent Statement:** Not applicable.

**Data Availability Statement:** The data presented in this study are available on request from the corresponding author.

**Acknowledgments:** The authors would like to acknowledge the factory personnel who helped and supported collecting relevant data for writing this paper. The authors did not receive any external funding for conducting this assessment.

**Conflicts of Interest:** The authors declare no conflict of interest.

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
