# Peer review of "An Assessment of Energy and Groundwater Consumption of Textile Dyeing Mills in Bangladesh and Minimization of Environmental Impacts via Long-Term Key Performance Indicators (KPI) Baseline"

_textiles, doi:10.3390/textiles2040029_

Round 1

Reviewer 1 Report

This article has analyzed the energy and groundwater consumption trend based on 15 textile dyeing mills in Bangladesh in 2019. The article is set up as a critical review of the failure criteria that in a guided way allows to select the most suitable criterion for the chosen case study. Long-term key performance indicators (KPI) reduction is set to a baseline by reducing energy and groundwater consumption in textile dyeing mills. The overall calculation can vary by the location of textile dyeing mills worldwide, the number of textile dyeing mills, and the timelines. This case study was limited to energy and groundwater consumption trends in textile dyeing mills in Bangladesh.

Reviewer 2 Report

A report for: textiles-1920033 An Assessment of Energy and Groundwater Consumption of Textile Dyeing Mills in Bangladesh and Minimize Environmental Impacts via Long-term KPI Baseline.

I have reviewed the manuscript that focused on the aanalysis of the energy and groundwater consumption trend based on textile dyeing mills in Bangladesh in 2019. This manuscript contains new and interesting data, but in my opinión needs some work to become suitable for publication.

 -Please improve the figure 1

 -Line 204. Data collected in each factory include annual Kg production, kWh electricity consumption, extracted groundwater in a million liters, and effluent discharged wastewater in a million liters. Without the need to offer many details, it is convenient to indicate the procedure followed

 -Table 2 Authors must explain why they select data from those territories and not others? And how do you get that data?

 -Figure 7 shows heavy metals released into the environment with effluent-treated wastewater from 15 textile dyeing mills. According to an effluent-treated wastewater analysis report from 15 textile dyeing mills, results detected nine heavy metals (boron, manganese, chromium, zinc, copper, nickel, cobalt, antimony, and lead). The question is: who analyzes these elements? and with what methods? What is the reliability of the results?

 -Line 353. I suggest to move the section 5. Limitations into section 4 Discusions

 - Part of the conclusions should be moved to the discussion section and in this way the extensive conclusions section is reduced.

 -Follow the rules of the journal especially the references section

 I wish those changes will contribute to improve your paper.

Round 2

Reviewer 2 Report

I went through the revised version of the manuscript and found that it had considerably improved from the original manuscript. The language and structure of the manuscript have improved and are more understandable. Therefore, the article is suitable for publication in its present form.